# Drivers of cadmium accumulation in *Theobroma cacao L*. beans: A quantitative synthesis of soil-plant relationships across the Cacao Belt

Jordon Wade[1]*, Marlon Ac-Pangan[2], Vitor R. Favoretto[3], Alexander J. Taylor[2], Nicki Engeseth[2], Andrew J. Margenot[3]*

1 School of Natural Resources, University of Missouri, Columbia, MO, United States of America,
2 Department of Food Science and Human Nutrition, University of Illinois Urbana-Champaign, Champaign, IL, United States of America, 3 Department of Crop Sciences, University of Illinois Urbana-Champaign, Champaign, IL, United States of America

* j.wade@missouri.edu (JW); margenot@illinois.edu (AJM)

**Data Availability Statement:** Data and accompanying code can be found at https://github.com/jordon-wade/Cacao-cadmium-meta-analysis.

## Abstract

Elevated cadmium (Cd) concentrations in cacao and cocoa-based products (e.g., chocolate) present a potentially serious human health risk. While recent regulatory changes have established a threshold of 0.8 mg kg$^{-1}$ for Cd content of cocoa-based products, the biophysical factors (e.g., climatic or edaphic conditions) that determine the amount of soil-derived Cd in the cacao bean are poorly understood and have yet to be quantitatively assessed across diverse production contexts. To determine the primary drivers of cacao bean Cd, we used the scientific literature to systematically compile a database of climatic, edaphic, and plant data from across the Cacao Belt, which is approximately 20 degrees latitude on either side of the equator. From this compiled dataset, we then used boosted regression trees to quantitatively synthesize and evaluate these drivers of cacao bean Cd. Total soil Cd concentration, soil pH, and leaf Cd were the best predictors of bean Cd content. Notably, we found that both available soil Cd and soil organic carbon (SOC) content had negligible effects on bean Cd. However, soil pH and SOC decreased the degree of bioconcentration of total soil Cd in the bean Cd concentration. Thus, given the difficulty in remediating soil Cd enriched soils, our results suggest that Cd mitigation strategies targeting plant physiology-based approaches (e.g., breeding, rootstocks) have a higher probability of success than soil-based strategies (e.g., remediation).

## Introduction

Cadmium (Cd) is a trace metal element of human health concern [1] that can cause a variety of adverse health outcomes, including endocrine disruption, osteoporosis [2], and has been classified as a carcinogen, especially in the context of Western diets [3]. Cadmium enters the anthropogenic trophic chain via soil-plant transfers [4–6]. Among dietary sources of Cd,

**Funding:** The author(s) received no specific funding for this work.

**Competing interests:** The authors have declared that no competing interests exist.

chocolate and other cocoa-containing food products (e.g. cocoa) are dominant exposure sources [7] and have received public and legislative attention in the past decade [8]. In particular, the European Union (EU)—which imports nearly half of the world's cacao production—recently defined a maximum permissible Cd content in cocoa products of 0.1–0.8 mg kg$^{-1}$, scaling with cocoa solids content [9]. The upper EU limit of 0.8 mg kg$^{-1}$ of high-cocoa foods (e.g., dark chocolate) has thus been used as a conservative threshold for seeds ("beans") of cacao (*Theobroma cacao*) [8, 10]. However, this threshold is often surpassed across much of the global Cacao Belt [11]. To protect human health, the EU and several countries (i.e., the United States) are increasing regulations on Cd content in cocoa-containing foods. These regulatory efforts decrease importer demand and scales the market value of cacao beans as a function of Cd content [8], with direct economic consequences for producers of this key tropical cash crop [12].

Cacao is the basis of livelihoods for over 8 million smallholder farmer households [13]. Incentivizing cacao production can support rainforest preservation and restoration efforts [14, 15] and offers a cash crop alternative to *Erythroxylum coca* (used to manufacture cocaine) [16]. Thus, the impacts of Cd regulations on cacao production are a serious constraint to its economic, ecological and even national security benefits. To resolve the Cd threat to the global cacao value chain, understanding soil-plant Cd transfers [17] across Cacao Belt production systems is a necessary first step to gauge the severity of Cd contamination and explain soil-based drivers of bean Cd uptake. Such knowledge ultimately stands to guide production practices that mitigate bean Cd, such as investing in soil Cd mitigation [10] versus germplasm- and horticulture (e.g., rootstock) based solutions [18].

To date, most evaluations linking soil Cd to bean Cd focus on specific regions, ultimately limiting insight into more generalized drivers of bean Cd content. Similar to other plants, cacao uptake of Cd is thought to reflect total soil Cd content and interactions with other edaphic properties (e.g. pH and cation-exchange capacity [CEC]), which impose well-characterized controls on Cd bioavailability in model soil systems [19]. However, the *in situ* interactions of these factors that drive Cd uptake and bean Cd accumulation remain poorly understood. Though the accumulation of Cd in the soil is thought to favor plant uptake [20–22], surveys at farm to national scales have found poor to moderate attribution of bean Cd to total soil Cd alone [23, 24]. Furthermore, Cd within the cacao plant can be remobilized and transferred from vegetative tissues such as leaves to beans [23], potentially confounding the relationship between soil Cd and bean Cd. Comprehensively evaluating soil(-leaf)-bean Cd relationships across the diversity of edaphic controls of Cd across the Cacao Belt can therefore capture variation in known soil drivers of Cd translocation to leaf and/or bean.

A first step towards addressing these issues is to quantitatively integrate our understanding of soil Cd in Cacao Belt soils while integrating the effects of both edaphic and climatic properties. Therefore, this systematic review and quantitative synthesis had two overarching objectives: 1) describe the characteristics of cacao-producing soils, and 2) describe edaphic and climatic drivers of bean Cd content in cacao production systems across the world. The ultimate goal is to identify and describe the multiple influences on cacao bean Cd concentration while also providing insight into broadly-applicable mitigation strategies. From these insights, we also aim to guide successful future research strategies.

## Materials and methods

### Systematic review and literature search

To synthesize current soil, leaf, and bean Cd literature, we conducted a literature search in September of 2020 on Web of Science (Thomson Reuters; www.webofknowledge.com) using

search terms "(("cadmium" OR "Cd" OR "metal*") AND ("cocoa" OR "cacao" OR "chocolate*" OR "Theobroma") AND ("soil*" OR "ground" OR "production" OR "farm*" OR "plantation*" OR "leaf" OR "leaves" OR "bean*" OR "nib*" OR "shell*"))". To ensure thorough literature coverage, we also conducted a similar search using the Scopus (Elsevier; www.scopus.com) and CAB Abstracts (CABI; www.cabi.org) databases in October of 2020. The specific sets of search terms for each can be found in the S1 File. These search terms resulted in a total of 1,113 records (Web of Science, n = 568; Scopus, n = 287; CAB Abstracts, n = 258). After removing duplicates, we screened a total of 785 abstracts, resulting in 457 full-text articles for assessment of eligibility requirements (Fig 1). We considered a study eligible if it either 1) measured soil Cd in cacao-based cropping systems or 2) measured both leaf and bean Cd from cacao. Inter-rater reliability was determined using a subset of 50 English language texts in two successive rounds of comparisons between the lead author and other authors. Inter-rater agreement was calculated using Cohen's kappa statistic, which ranged from 0.73 to 1.00, considered "substantial" to "almost perfect" agreement [25, 26]. Authors then screened non-English texts for eligibility, including Spanish, Portuguese, and French language texts. Additionally, we contacted authors for datasets that were incomplete or only partially available in the publications.

## Data extraction and aggregation

Of the 67 studies that met the eligibility requirements, 31 studies had usable data. Reasons for unusable data included: only reporting the grand mean, no reports of variability (e.g., standard deviation or standard error), ambiguously defined sample sizes, or reporting of data at varying hierarchical levels (e.g. reporting soil data at the field level and bean data at the regional level). Using the Critical Appraisal Tool developed by the Collaboration for Environmental Evidence [27], we used a total of eight criteria to determine the overall quality and risk-of-bias of each study (S2 Table in S1 File). The 31 studies with usable data contained a total of 489 site-years of data comprising a total of 2,127 total observations from 10 countries. We extracted 1,122 individual data points from these studies, which we used as the final dataset for our analyses. In 16 of these 31 studies, we extracted individual observations or obtained original observations from authors, so each extracted value represented a single sample. In the remaining studies, extracted data points were reported as the mean of multiple measurements, resulting in an extracted value representing anywhere from 3 to 77 individual observations (mean = 10.7, median = 4). When the data was present and in a usable format, soil, leaf, and bean Cd data were extracted. Nearly all studies did not differentiate between just the bean (i.e., nib) and whole bean (i.e., shell + nib) in the methods section, although shells tend to have higher Cd content than nibs [22, 28, 29]. Here we assume that the lack of differentiation between shell and nib implies the whole bean was processed and analyzed for Cd together, rather than analyzed separately. We will use the term "bean" rather than "whole bean" to reflect this uncertainty. Wherever possible, additional soil characterization data—such as pH, soil texture, soil organic carbon (SOC) content, and CEC—were co-extracted with soil, leaf, and bean Cd data. Soil pH data is expressed as pH measured in water, with a conversion from pH measured in background solutions of KCl [30] or $CaCl_2$ [31] applied as necessary. Similarly, SOC content was the preferred measure of organic matter and conversions from organic matter measured via loss-on-ignition using the factor of 1.74 were applied as necessary [32]. Due to the generally acidic soil pH values, we interpreted total C as total SOC [33, 34]. Wherever possible, each site's GPS coordinates were extracted and used to estimate mean annual precipitation (MAP) and temperature (MAT) from the WorldClim database [35, 36]. Reliable extraction of continuous data such as soil, leaf, and bean Cd concentrations or GPS coordinates on a map image was performed using the WebPlotDigitizer [37, 38], the values of which were subsequently

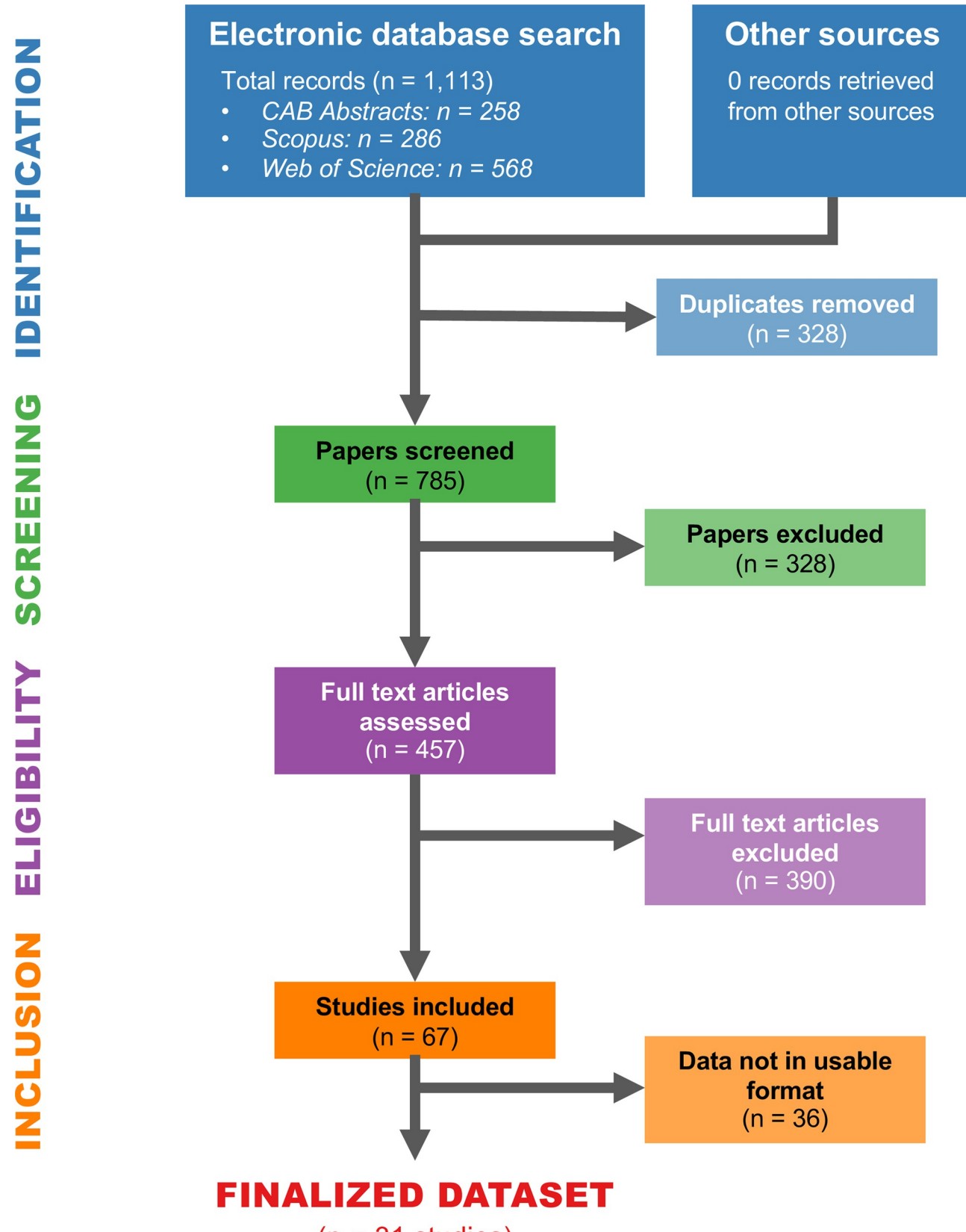

**Fig 1. PRISMA flow diagram illustrating study selection and exclusion process for each screening step.**

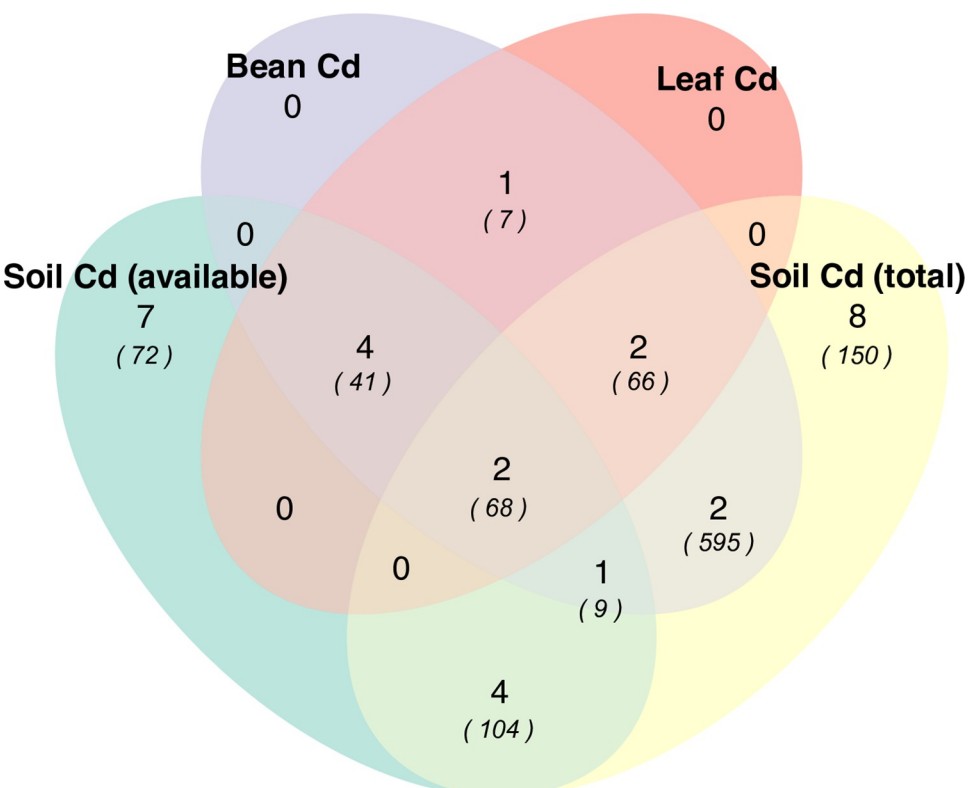

**Fig 2. Venn diagram of number of studies containing each type of cadmium data: Soil available Cd, total soil Cd, leaf Cd, or bean Cd concentrations.** Values in parentheses are the number of observations extracted from the studies.

verified by a secondary extraction. To determine the overall Cd accumulation tendencies of cacao, we calculated bioconcentration factors (BCF)—which is also referred to as transfer factor or bioaccumulation factor—for observations that had detectable values of both total soil Cd and whole bean Cd. BCF was calculated as the ratio of bean Cd to total soil Cd concentrations, where values > 1 indicate bioaccumulation of Cd in the beans (relative to the soil) and values < 1 indicate dilution of Cd in the beans (relative to the soil) [39].

Studies varied widely in the type of reported soil and plant Cd data (Fig 2). A total of 11 out of 31 studies (35%) reported some combination of soil, leaf, and bean Cd data, 19 (61%) reported only soil Cd data, and just 1 study contained only paired leaf-bean Cd data (3%). While the amount of supporting edaphic and geographic data was mixed, 29 studies (94%) reported at least one soil characterization variable and 29 studies (94%) had viable geolocation data to allow for estimates of MAP and MAT. More information on the geographic distribution of variables and methods used to measure soil Cd are in S3 and S4 Tables in S1 File, respectively.

## Data analysis: Gradient boosted regression trees

Boosted regression trees (BRTs) integrate the flexibility of classification and regression trees (i.e., decision trees) with the "learning" capabilities of machine learning. Decision trees iteratively search for "splits" within the data to determine thresholds of effects of independent variables on the dependent variable. However, this recursive partitioning of decision trees can overfit the model to the data. One solution to the problem of overfitting is to have a model that "learns" from previous models by both resampling and accounts for observations in previous

models that were poorly described. BRTs use a "boosting" strategy to learn from previous models by fitting a succession of shallow decision trees to develop a single, combined model [40–42]. Essentially, BRTs create one "rule of thumb" from many "rules of thumb" [40]. The ability to model nonlinear relationships—specifically the ability to account for interactions between multiple nonlinear variables—and deal with missing data [43] are distinct advantages of BRTs. We used BRTs to estimate cacao bean Cd concentrations, using soil Cd concentrations (available and total), MAT, MAP, soil pH, CEC, SOC, clay content, and leaf Cd concentrations as driving variables. Where soil Cd concentration was available for multiple soil depths, we calculated a depth-weighted soil Cd concentration (assuming a constant bulk density). Similarly, we used BRTs to estimate the BCF of Cd in cacao, using depth-weight total soil Cd, MAT, MAP, soil pH, CEC, SOC, clay content, and leaf Cd concentration.

To optimize our BRT models, we used a range of tuning parameters for our model's learning rate, the number of splits to be used in each subsequent decision tree, the minimum number of observations per node of the tree, and the subsampling rate from our training set (80% of the total dataset). The specifics of these tuning parameters can be found in S5 Table in S1 File. We used 2,000 decision trees to estimate the model error for each of the 10,098 permutations of our tuning parameters, selecting the model with the lowest root-mean-square error (RMSE) as the final model. We used the permutation test to determine overall predictor variable importance, where the change in model error due to the inclusion/exclusion of a variable is proportionate to its importance. We used partial dependence plots to visualize the effects of predictor variables on bean Cd concentration in our BRT models. Partial dependence plots calculate the average marginal effects of the predictor variables on the response variable while holding all other effects constant. Statistical analyses were run using the *gbm* [44], *rsample* [45], *caret* [46], and *xgboost* [47] packages in RStudio [48] and the *pdp* package [49] was used in conjunction with the *ggplot2* package [50] to construct partial dependence plots. Other basic statistical analyses and data manipulation used the *dplyr* [51] package and the *ggpubr* [52] package to construct visualizations. All maps were made in *maps* R package [53] which use Natural Earth imagery (public domain).

## Results & discussion

### Geographic, climatic, and edaphic distribution of studies on cacao Cd content

Of the 31 studies that fit our inclusion criteria, the majority of the studies (n = 22, 71%) were conducted in Central and South America, with the remainder from West Africa (n = 7, 23%) and Southeast Asia (n = 2, 6%). The total number of observations is similarly biased towards Central and South America (Fig 3A–3C). While the proportion of studies from Southeast Asia approximately reflect this region's proportion of total global cacao production (~6%), Central and South America, driven by observations from Ecuador, are overrepresented relative to their portion of global production (~18%), whereas West Africa is underrepresented relative to their contribution to global production (~75%) [54, 55]. Notably, Côte d'Ivoire—which produces 40–45% of the world's cacao—is absent from the literature. The relative dearth of data from West Africa highlights an acute need for additional work in the region to better understand cacao Cd dynamics. It is possible that the disproportionate number of studies in Central and South America reflects the differences between higher grade Ecuadorian "fine" cacao and West African "bulk" cacao, as fine cacao is used for dark chocolates with high cacao solid contents and thus more stringent regulatory limits on Cd concentrations [56, 57].

Despite the geographic bias of the dataset towards Central and South America (Fig 3A–3C), the distribution of climatic variables encompasses growing conditions thought to be ideal for

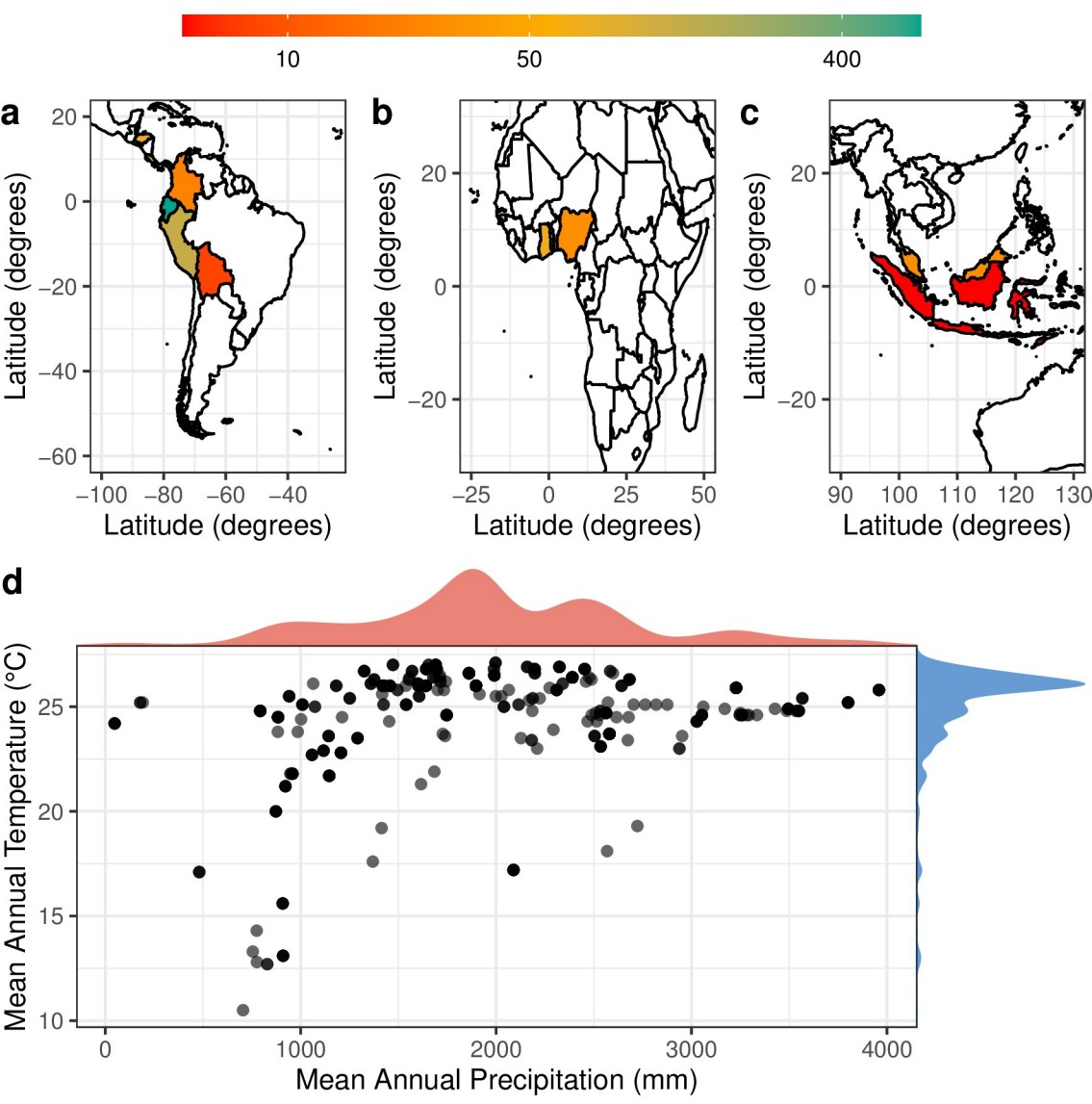

**Fig 3.** Number of observations within (a) South America, (b) Africa, and (c) southeast Asia, as well as (d) the distribution of the mean annual temperature (MAT) and mean annual precipitation (MAP) from those sites. Not all studies included enough information to reliably obtain MAP and MAT estimates.

cacao. Although the full range of MAT in our dataset is from 10.5 to 27.1˚C (Fig 3D; mean = 25.0˚C), nearly all of the values (90%) were between 21.2 and 26.8˚C, which is comparable to the typical range of cacao production from 22.4 to 26.7˚C [58]. The range of MAP values was skewed slightly higher than is conventional for cacao production (Fig 3D). Generally, annual rainfall below 1400 mm and above 2500 mm is not considered ideal for cacao production [58], but only ~50% of our values fell within this range (Q1 = 1605 mm, Q3 = 2390 mm).

## Depth distribution of soil Cd

The overwhelming majority of studies of both available and total soil Cd took soil samples from within the 0–30 cm depth (Fig 4A, 4C and 4E). This is consistent with a surficial bias across soil science studies more broadly [59]. Both available and total Cd showed a similar

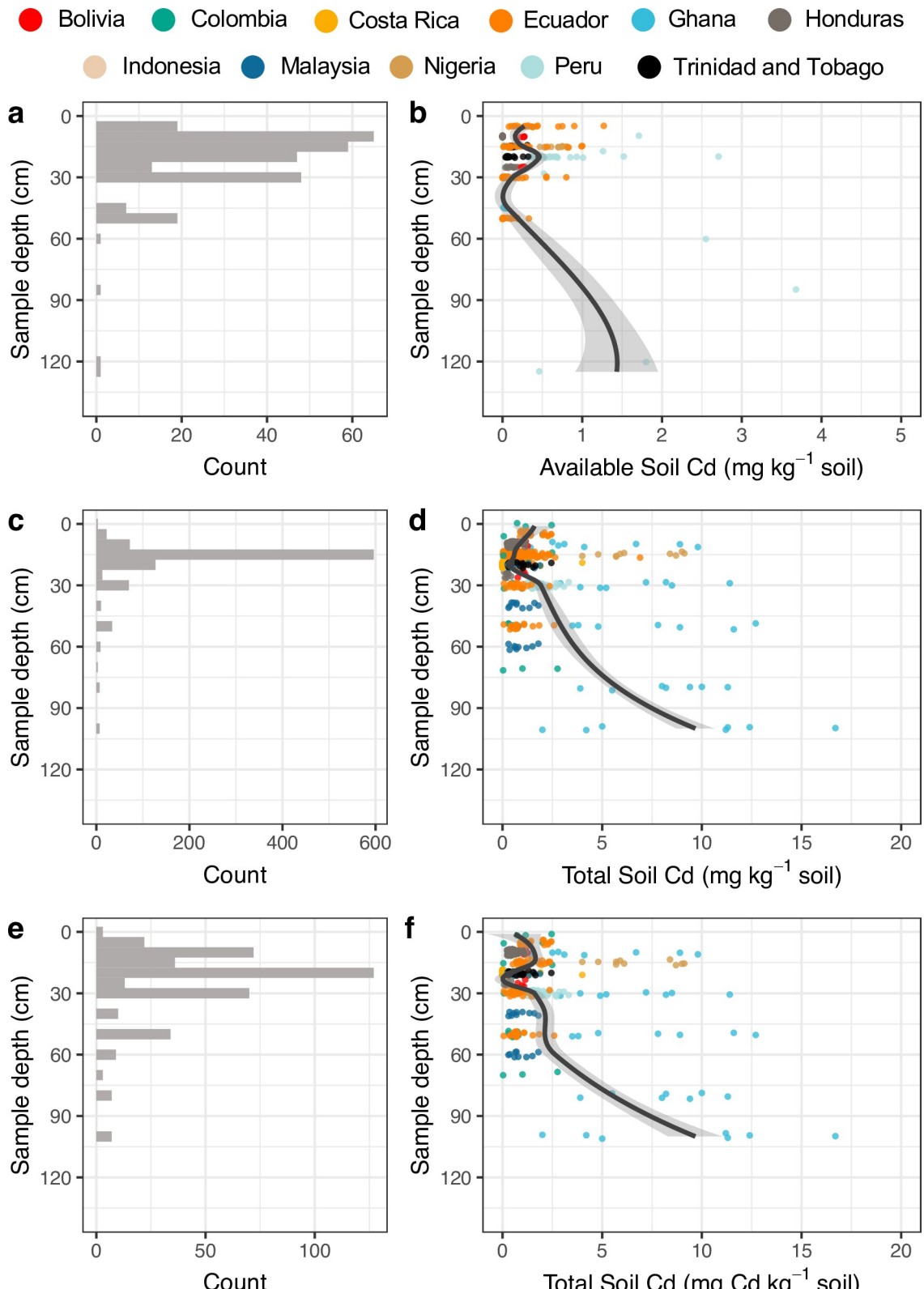

**Fig 4.** Distribution of (a,c,e) the number of observations by sample depth and (b,d,f) the Cd concentration of those samples for both (a,b) available and (c,d,e,f) total soil Cd. Due to the high proportion of our total observations coming from a single study from Ecuador [24], total Cd distributions both (c,d) with and (e,f) without that study have been shown. Shaded regions indicate 95% confidence intervals of fitted curve.

trend of decreasing soil Cd from the surface to 25 to 35 cm in depth, followed by a marked increase at greater depths (Fig 4B, 4D and 4F). While available Cd was lowest at ~35 cm depth (Fig 4B), total Cd had a slightly shallower minima at ~25 cm depth. Available soil Cd at shallower depths tended to be < 1 mg Cd $kg^{-1}$ soil whereas total Cd tended to be < 3 mg Cd $kg^{-1}$ soil. In the subsurface, available soil Cd increased by $\approx$ 50% and total Cd increased by $\approx$ 200%. Although we did not recover many published observations from depths > 40 cm, consistent with generally surface-biased sampling depth (0–27 cm; ref) the increase in both available and total Cd suggests that the dominance of surface sampling soils (< 30 cm) to determine Cd may substantially underestimate how soil Cd is available for cacao uptake. Although most of cacao's lateral roots are found within the top 0 to 20 cm of soil, the taproots can extend well below 1 m in depth [60]. Currently, there is substantial uncertainty around the location within the soil profile that plants are most likely to take up Cd [17], raising the possibility that surface-biased soil sampling could be misrepresenting the nature of plant-soil Cd relationships.

## Primary drivers of cacao bean Cd content

Bean Cd was driven by total soil Cd and soil pH (Fig 5A). Total Cd and pH of soils are well-known predictors of soil solution Cd, both for annual crops [61] and for cacao [24]. Total Soil Cd generally had positive effects on bean Cd, although it plateaued at ~1.5 mg Cd $kg^{-1}$ soil. (Fig 5B). While previous work has shown total soil Cd to increase bean Cd, they did not report a plateaued effect, which could be obscured by their use of a double-log transformation [24]. The combined influence of total Cd and pH in this dataset is consistent with these being

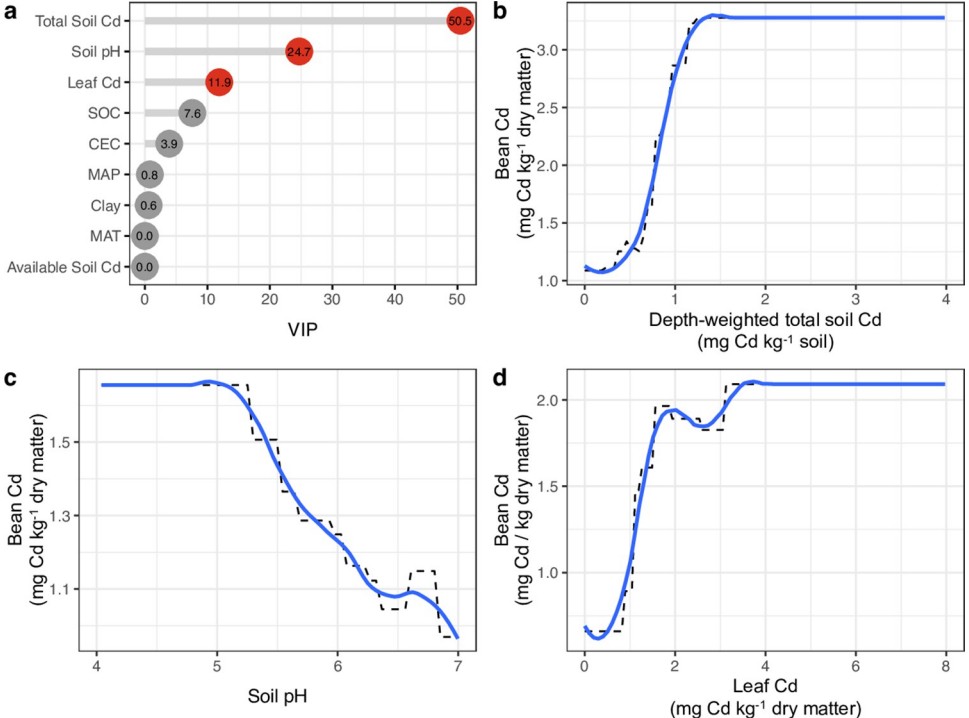

**Fig 5. Drivers of cacao bean Cd levels and their modeled relationships.** Boosted regression trees (BRTs) reveal (a) the relative influence of variables in predicting bean Cd content and (b-d) the partial dependence plots associated with their marginal effects. Blue lines indicate a smoothed line for ease of interpretation while dashed lines are derived average marginal effects. Red dots in (a) denote a significant contribution to cacao bean Cd content that are explicitly visualized in (b-d).

dominant drivers of Cd solubility, and subsequently bioavailability. The amount of Cd in soil solution is strongly dictated by pH [19, 62], and mediates how much of the total Cd pool is present as soluble $Cd^{2+}$ (i.e. the partitioning coefficient). As predicted by chemical theory, the bioavailability of soil Cd is greater at lower pH, and strongly diminished at higher pH (Fig 5C). However, the decline in soil Cd bioavailability (as reflected by bean Cd) observed in our dataset initiates at a lower pH than model systems indicate (e.g., Cd-$CO_3$ at pH 7) [63], ultimately suggesting that *in situ* dynamics are not fully captured by these model systems. Additionally, the inconsistent effects of soil pH on bean Cd at higher pH (S4B Fig in S1 File) indicate that Cd uptake in these ranges is likely more complex than at acidic pH values. Unlike soil pH, total Cd is more site-invariant (S4A Fig in S1 File). One explanation for more variable soil pH effect at higher pH (> 5.5–6.0) is the potential confounding presence of calcium (Ca). In particular, soil Ca is antagonistic to soil Cd uptake [64] and may also slow Cd transport from roots to leaves [65].

Notably, soil available Cd was not identified as a significant driver of bean Cd (Fig 5A), suggesting that these measures did not appear to be plant-available. Available soil Cd is commonly measured as salt- and/or acid-extractable Cd [8]. These extractions are considered plant-available pools [4, 8], despite the wide range in Cd concentrations isolated by these extraction methods. While our dataset integrated a plurality of soil methods assumed to provide insights on "available" soil Cd (S4 Table in S1 File), two related lines of reasoning suggest this was not a confounding factor: 1) the close correlation between available Cd and total Cd regardless of method (S1 Fig in S1 File) and 2) the lack of explanatory power of available Cd even when total soil Cd was removed from the model (S6 Table in S1 File). Thus, available Cd offers little information about bean Cd concentration independent of its covariation with total Cd. Previous work in ryegrass using various methods has similarly found that "available" soil Cd did not represent the portion taken up by plants [66]. In contrast to available Cd determined by extractions, exchangeable Cd is the fraction of total Cd that is potentially available for plant uptake [67]. This implicates CEC as a relevant, albeit minor soil variable, which is borne out here (Fig 5A, S6 Table in S1 File) and supported more generally by the control of CEC on $Cd^{2+}$ in soil solution [68, 69]. Sequential extraction methods are often used to separate the total Cd pool into standardized fractions defined by chemical solubility [4], which in some cases approximate conceptual pools such as bioavailable or parent material Cd [70, 71]. However, the overall lack of importance of estimates of "available" soil Cd (Fig 5A) suggests that these labor-intensive measures offer little insight into cacao Cd uptake.

Unexpectedly, we found a minor effect of SOC on bean Cd (Fig 5A, S6 Table, S3 Fig in S1 File). The general influence of organic matter on metal element bioavailability [72] in soils is mediated by ligand exchange with carboxylate moieties, which has been well detailed for Cd [19, 62]. Though this partitioning between soluble and organically-associated Cd does not appear to inform cacao bean Cd content, our model provides some evidence of its occurrence: at lower concentrations (~1%), SOC had a positive effect on bean Cd (S3 Fig in S1 File), but this effect was negligible at 2.5–3% SOC. Therefore, while the overall effect of SOC was small compared to other factors, the relative absence of enough SOC (i.e. at negligible SOC contents) may have had a greater effect on Cd availability than in the presence of high SOC. Our use of total SOC content neglects the potential role that SOC composition can have on Cd bioavailability [73]. However, there is currently limited *in situ* evidence of organic amendments altering cacao Cd content. Studies investigating the effect of carboxylate-rich amendments either translate poorly from greenhouse to field evaluations of plant Cd uptake [74] or infer soil Cd availability through extractions [4], which we found to be poor predictors of bean Cd (Fig 5A). Thus, propositions of increasing overall SOC to immobilize soil Cd [4, 74, 75] are not likely to be successful.

Though not commonly measured (S3 Table in S1 File), leaf Cd was an unexpected driver of bean Cd content (Fig 5A). Although this could reflect a leaf-total soil Cd relationship (S1 Fig in S1 File), the exclusion of leaf Cd nearly doubled the model estimation error (S6 Table in S1 File), suggesting that leaf Cd is imparting unique and useful information about bean Cd. The high relative importance of leaf Cd could reflect increasing evidence of cultivar-specific Cd uptake and accumulation tendencies [76] that complicate relationships of bean Cd with soil Cd [23, 29, 77]. For example, 13-fold differences in bean Cd were found among cacao cultivars grown in the same soil [29]. Variation in Cd translocation from soil to root via transporter proteins and from root to leaves via heavy metal ATPases can be strongly influenced by cultivar genetic diversity [78]. Bean Cd isotopic fractionation ($\delta^{114/110}$Cd) further suggests that cultivars may acquire Cd from distinct soil pools [77], although our results suggest these pools are not well-approximated by common chemical extractions of "available" Cd (Fig 5A). However, accounting for cultivar type did not explain bean Cd concentrations nor soil-bean Cd relationships in an Ecuador-wide survey [24]. Cultivar-specific tendencies are a relatively underexplored area of cacao Cd dynamics. This literature review initially targeted this data for inclusion, but the overwhelming majority of studies either did not denote the cultivar of cacao or only reported mixtures of cultivars.

## Bioconcentration of Cd in cacao beans

Cacao tended to be a bioaccumulator of Cd in the beans. A total of 77.0% of observations had BCF values > 1, indicating that the concentration of Cd in the cacao beans was higher than the concentration of total Cd in the soil (Fig 6A). Annual crops such as rice, maize, and wheat tend to have BCF values < 1 [79, 80], whereas perennials—including cacao—tend to have BCF values > 1 [39, 81]. While most of the observations had BCF values between 1 and 5 (66.2%), BCF values reached as high as 30 and as low as 0.1 (mean BCF = 2.5, median BCF = 1.8), indicating a wide range of potential bioaccumulation in cacao beans. This wide range of BCF values reflects the diversity of Cd uptake, translocation, and accumulation in beans amongst cultivars [23, 24, 29, 76, 77, 82].

Soil pH and SOC were the primary drivers of BCF and had generally negative, albeit nonlinear relationships with BCF (Fig 6B and 6C). These nonlinear effects of soil pH and SOC were similar to their effects on bean Cd (Fig 5C and S3A Fig in S1 File); increasing acidic soil pH from pH = 5 to pH = 7 and increasing a low SOC content soil from 2% to > 3.5% would cut the BCF in half (Fig 6B and 6C). This suggests that the effects of soil pH and SOC is primarily a result of their effects on bean Cd. However, both total soil Cd and leaf Cd had a relatively minor effect on BCF (< 13% and < 6% reduction in error, respectively; S6 Table in S1 File). Thus, while total soil Cd content is a primary driver of bean Cd (Fig 5A and 5B), management strategies can still be effective in the degree to which the bean Cd concentration reflects the total soil Cd concentration.

## Current data gaps and recommendations

Our study identifies geographic and methodological data gaps that can be readily addressed in future work. Geographically, there is a severe under-evaluation of West African cacao systems given its dominance of global cacao exports. This makes disentangling the interactive effects of biogeochemistry and climate quantitatively challenging and somewhat uncertain. Importantly, this data gap constrains the applicability of our findings to the "fine" cacao typically produced in Central and South America. While the primary driving variables identified here should theoretically apply beyond this region, we cannot speak to these underrepresented regions with a high degree of confidence. Methodologically, future work should differentiate between

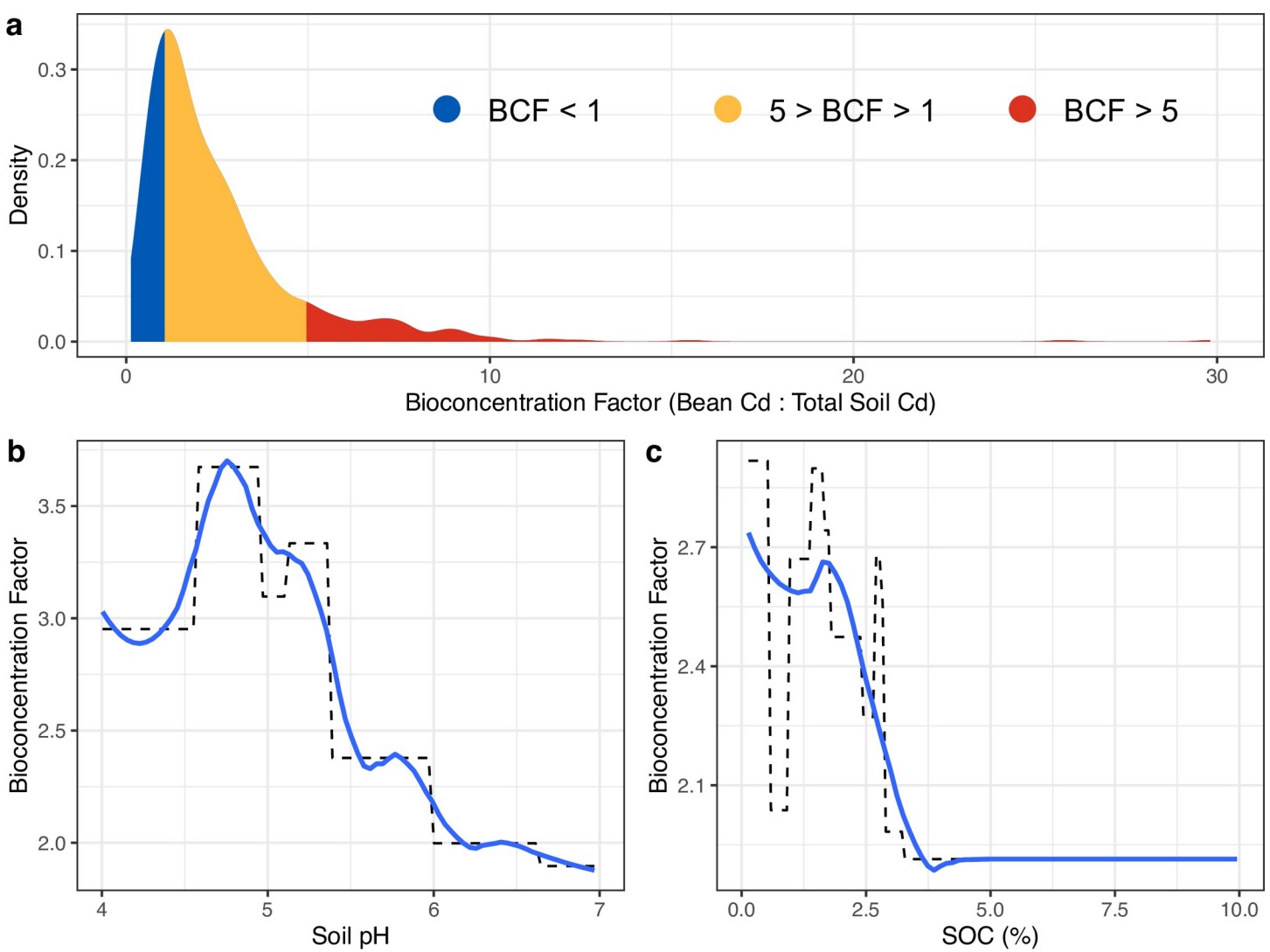

**Fig 6.** The (a) distribution and (b and c) primary drivers of cacao bioconcentration factors (BCF). BCF values > 1 indicate an accumulation of Cd in cacao beans, relative to the concentration of total Cd in soil. Modeled marginal effects of (b) soil pH and (c) soil organic carbon (SOC) on BCF are shown using partial dependence plots. Blue lines indicate a smoothed line for ease of interpretation while dashed lines are derived average marginal effects.

analyses conducted on the whole bean (shell + nib) and analyses conducted solely on the nib. The nib is the part of the bean that is used in food products and is likely lower in Cd than the whole bean [22]. Therefore, nib Cd concentration is likely the most germane to analyses and regulation of cacao Cd content and human health. Additionally, the tendency towards shallow soil sampling represents another substantial methodologically-based data gap. It is not yet clear which soil depths are the most significant for Cd accumulation, depending on the source, and plant Cd uptake [17], nor how nutrient and water stress may shift root allocation and thus potentially the depth of soil Cd uptake. Though the majority of cacao feeder roots are generally concentrated at 0–20 cm depth, the taproot can easily extend to 150 cm depth with appreciable feeder roots extending to 30 cm and, in mature plants, up to 120–150 cm [60]. In light of this current data gap, soils should be sampled beyond surface (0–30 cm) depths to account for sub-surface Cd that can contribute to cacao uptake. A more effective sampling scheme would measure total soil Cd at multiple, deeper soil depths instead of measuring both extractable and

total soil Cd in shallower depths. In addition to deeper soil sampling, leaf sampling can provide more information than widely used measures of soil available Cd and should be included in future studies. Here, more mechanistic information is needed to understand the nature of leaf-bean Cd relationships, including cultivar-specific Cd remobilization among these tissues [77, 83]. To this additional data gap, studies should report cultivar information, including differentiating between rootstock and scion when applicable [24].

## Synthesis and conclusion

This work identifies the primary soil and plant drivers of bean Cd and bioconcentration, ultimately providing several key insights and opportunities to address the Cd threat to the global cacao economy. While the accuracy of this work (RMSE = 0.636 mg Cd kg$^{-1}$ bean dry mass; S6 Table in S1 File) was not sufficient to meet regulatory limits for predictive accuracy ($\approx$ 0.1 mg Cd kg$^{-1}$ bean dry mass), it offers high-level guidance on which mitigation strategies are likely to be (cost) effective. First, given the difficulty and expense of soil Cd remediation [84], the finding that total Cd in soil, rather than available Cd, is the single greatest driver of bean Cd suggests other more effective and affordable avenues for cacao Cd mitigation than remediation. Our findings suggest that the most viable soil-based mitigation strategy is to elevate soil pH by liming. Increasing soil pH to pH > 6.0 not only decreases bean Cd, but decreases the degree of biomagnification of total soil Cd in the beans. Liming can chemically immobilize Cd and thus decrease availability to the cacao plant, while also providing co-benefits to cacao productivity by ameliorating acidity [85]. For example, a one-time application of lime to soils decreased cacao uptake of Cd by up to 1.7-fold [10]. Thus, the effect of liming can decrease bean Cd directly by decreasing uptake and indirectly by increasing productivity (i.e. dilution). There is also evidence that calcium (Ca) may inhibit Cd translocation in other tree crops [65], suggesting that specifically using calcitic lime to raise soil pH could have synergistic effects on cacao Cd uptake. Fully separating these effects would be useful to inform pH- based mitigation strategies of soil Cd. The heterogeneous effects of soil pH on bean Cd, particularly at pH > 6.0, identified in our study raise the possibility of other factors interacting with pH at circumneutral to alkaline values. The inexplicable heterogeneity represents a substantial gap in mechanistic understanding of soil-cacao plant Cd transfers. While studies have found mixed effects of cultivar on Cd uptake [23, 24, 76], this heterogeneity also presents a substantial mitigation opportunity. For example, there is evidence from other crops that intentional pairings of rootstock with soil pH (i.e. leveraging genotype x environment interactions) provide synergistic opportunities for mitigating Cd uptake and bioconcentration [86, 87].

Collectively, our results demonstrate that the effect of relatively static soil properties (e.g., total soil Cd) on absolute levels of bean Cd are significant, whereas more malleable factors (e.g. soil pH or SOC) have relatively smaller effects. However, the malleable factors of soil pH and SOC can influence the extent to which the total soil Cd is concentrated in the cacao beans. Therefore, while soil based strategies (e.g. liming or increasing SOC content) may help mitigate bean Cd accumulation, the prohibitive cost of decreasing total soil Cd suggests that future efforts to decrease cacao bean Cd concentrations may be best pursued through breeding [18, 76] or other such plant physiologically-based strategies.

## Supporting information

**S1 Checklist. 2020 PRISMA checklist.**
(DOCX)

**S1 File.**
(DOCX)

**S1 Graphical abstract.**
(PDF)

# Acknowledgments

The authors would like to thank Dana Dubinski for her contributions to earlier versions of this work.

# Author Contributions

**Conceptualization:** Marlon Ac-Pangan, Vitor R. Favoretto, Alexander J. Taylor, Nicki Engeseth, Andrew J. Margenot.

**Data curation:** Jordon Wade, Marlon Ac-Pangan, Vitor R. Favoretto, Alexander J. Taylor, Andrew J. Margenot.

**Formal analysis:** Jordon Wade.

**Investigation:** Jordon Wade, Marlon Ac-Pangan, Vitor R. Favoretto, Alexander J. Taylor, Nicki Engeseth, Andrew J. Margenot.

**Methodology:** Jordon Wade, Marlon Ac-Pangan, Vitor R. Favoretto, Alexander J. Taylor, Andrew J. Margenot.

**Project administration:** Jordon Wade, Nicki Engeseth, Andrew J. Margenot.

**Writing – original draft:** Jordon Wade, Andrew J. Margenot.

**Writing – review & editing:** Marlon Ac-Pangan, Vitor R. Favoretto, Alexander J. Taylor, Nicki Engeseth.

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
