## [Decision Letter · Decision Letter 0]

29 Jul 2021

PONE-D-21-13459

Drivers of cadmium accumulation in Theobroma cacao L. beans: a meta-analysis of soil-plant relationships across the Cacao Belt

PLOS ONE

Dear Dr. Wade,

Thank you for submitting your manuscript to PLOS ONE. After careful consideration, we feel that it has merit but does not fully meet PLOS ONE’s publication criteria as it currently stands. Therefore, we invite you to submit a revised version of the manuscript that addresses the points raised during the review process.

We look forward to receiving your revised manuscript.

Kind regards,

Muhammad Rizwan

Academic Editor

PLOS ONE

2. Please ensure that you include a title page within your main document. You should list all authors and all affiliations as per our author instructions and clearly indicate the corresponding author.

3. We note that Figure 3, in your submission contain [map/satellite] images which may be copyrighted. All PLOS content is published under the Creative Commons Attribution License (CC BY 4.0), which means that the manuscript, images, and Supporting Information files will be freely available online, and any third party is permitted to access, download, copy, distribute, and use these materials in any way, even commercially, with proper attribution. For these reasons, we cannot publish previously copyrighted maps or satellite images created using proprietary data, such as Google software (Google Maps, Street View, and Earth). For more information, see our copyright guidelines: http://journals.plos.org/plosone/s/licenses-and-copyright.

a. You may seek permission from the original copyright holder of Figure 3 to publish the content specifically under the CC BY 4.0 license. 

4. We note that this manuscript is a systematic review or meta-analysis; our author guidelines therefore require that you use PRISMA guidance to help improve reporting quality of this type of study. Please upload copies of the completed PRISMA checklist as Supporting Information with a file name “PRISMA checklist”.

Additional Editor Comments (if provided):

Reviewers' comments:

Reviewer's Responses to Questions

**Comments to the Author**

1. Is the manuscript technically sound, and do the data support the conclusions?

Reviewer #1: Yes

Reviewer #2: Yes

2. Has the statistical analysis been performed appropriately and rigorously? 

Reviewer #1: Yes

Reviewer #2: Yes

3. Have the authors made all data underlying the findings in their manuscript fully available?

Reviewer #1: Yes

Reviewer #2: Yes

4. Is the manuscript presented in an intelligible fashion and written in standard English?

Reviewer #1: Yes

Reviewer #2: Yes

5. Review Comments to the Author

Reviewer #1: Comments to the Author

Authors have attempted a systematic review and quantitative synthesis for the detailed description of the characteristics of cacao-producing soils and edaphic and climatic drivers of bean Cd content in cacao production systems. They have identified and described multiple effects on cacao bean Cd concentration. They have also provided insight into broadly applicable mitigation strategies which can guide future research strategies.

This review manuscript is well written, however, there are some minor mistakes that must be resolved before publication.

Abstract:

Line 12, 13. The authors mentioned that elevated cadmium (Cd) concentrations in cacao and cocoa-based products present a potentially serious human health risk. What kind of health risks? Better to give detailed risks factors related to elevating the Cd concentration in the introduction section.

Line 19: Cacao belt referred to? The authors need to elaborate.

Introduction

Line 29: INTRODUCTION. (Only the first letter will be of uppercase followed by lowercase. Follow the same for other headings as well throughout the manuscript)

Line 46; replace “an” with “a”

Line 55: focus (focuses)

Line 61: remains (remain)

Line 62: at farm (at the farm)

Line 66: soil(-leaf)-bean (provide space)

Line 75: on cacao (on the cacao)

Line 76: From these insights (From these insights,) put a comma.

• Provide updated references for health risks associated with elevated concentration of Cd in Cacao and related products.

• Provide some updated references from similar studies in the introduction section

Material and Methods:

Line 81: on Web of Science (on the Web of Sciences)

Line 82-84: Check whether these terms will be italic or normal style?

Line 96: agreement[23,24] (create space)

Line 126: were applied (was applied)

Line 129: to estimate mean (to estimate the mean)

Line 132: GPS coordinates on a map image was performed (GPS coordinates on a map image were performed)

Line 144: characterization variable (characterization variables)

Line 146: (respectively). Remove the brackets.

Line 150: More detail is needed for Boosted regression trees (BRTs)

Line 175: RStudio[46] (create space)

Results & Discussion

Line 199: (Figure 3a, 3b,and 3c), (remove the comma 3b, and create space)

Line 201: 27.1ºC (create space)

Line 203: 7 ºC[55]. (create space)

Line 220: Although most of cacao’s (Although most of the cacao’s)

Line 231-233: Rewrite the sentence

Line 249: pools[2,6], (create space)

Line 277-278: rewrite the sentence and create space.

Line 292: in bean Cd were found (in bean Cd was found)

Line 362: suggest (suggests)

Conclusion:

Include a comprehensive and concise conclusion

References

• The authors need to double-check the references both in the text and bibliography

• Name of the journal should be in abbreviated form

• Check all references for journal references formate

General comments:

• Overall, the manuscript needs to be revised with updated references and clear methodology with proper references.

• The authors also need to check the whole manuscript for grammar, spelling and spacing mistakes to be corrected.

• The text should be in Times New Roman.

• The Pont size of the main headings (Title, Abstract, Introduction, Materials and Methods, Results and Discussion, Conclusion etc.) should be 18 according to journal formate

• Subheading Pont size should be 16

• The authors need to follow the journal formate for type texting.

Reviewer #2: Comments for authors

The current manuscript has been structured and written well i.e. “Drivers of cadmium accumulation in Theobroma cacao L. beans: a quantitative synthesis of soil-plant relationships across the Cacao Belt”. However, I have few following concerns that authors should incorporate to improve their manuscript:

Abstract:

Line 14. Mention threshold limit for cadmium here.

Line 15. Provide names of few biophysical factors.

Results relevant to BCF should be mentioned in the abstract.

Line 26. Which soil-based strategies? Mention few ones.

I will recommend increasing the keywords up to 5 numbers.

Introduction:

Line 32. “other cocoa-containing food products are dominant exposure sources.” Which other food products? Provide please.

The overall formatting of the article needs to be corrected. For example, the paragraph justification in introduction section. Please check in whole manuscript and correct it.

Lines 39-41. “To protect human health, the EU and several countries are increasing regulations on Cd content in cocoa-containing foods.” Which other several countries? Please write names of those countries.

Line 46. “and offers an cash crop alternative to Erythroxylum coca.” Remove ‘an’.

Materials and Methods:

Methodology section is good.

Result and Discussion:

Results and discussion section is presented and discussed nicely. However, in axis of figures in some places, authors have used ‘mg/kg’ and in other places, it is ‘mg kg-1. I will recommend using ‘mg kg-1’. Please check in all figures and in whole manuscript and correct it.

Synthesis and future work:

Lines 383-386. Please revise these lines and write in a more comprehensive way.

References:

References should be according to according to journal given format.

6. PLOS authors have the option to publish the peer review history of their article (what does this mean?). If published, this will include your full peer review and any attached files.

Reviewer #1: No

Reviewer #2: **Yes: **Afzal Hussain

---

## [Author Response · Author response to Decision Letter 0]

22 Nov 2021

Please see attached Response to Reviewers file.

---

## [Decision Letter · Decision Letter 1]

15 Dec 2021

Drivers of cadmium accumulation in Theobroma cacao L. beans: a quantitative synthesis of soil-plant relationships across the Cacao Belt

PONE-D-21-13459R1

Dear Dr. Wade,

We’re pleased to inform you that your manuscript has been judged scientifically suitable for publication and will be formally accepted for publication once it meets all outstanding technical requirements.

Kind regards,

Muhammad Rizwan

Academic Editor

PLOS ONE

Additional Editor Comments (optional):

Reviewers' comments:

Reviewer's Responses to Questions

**Comments to the Author**

1. If the authors have adequately addressed your comments raised in a previous round of review and you feel that this manuscript is now acceptable for publication, you may indicate that here to bypass the “Comments to the Author” section, enter your conflict of interest statement in the “Confidential to Editor” section, and submit your "Accept" recommendation.

Reviewer #1: All comments have been addressed

Reviewer #2: All comments have been addressed

2. Is the manuscript technically sound, and do the data support the conclusions?

Reviewer #1: Yes

Reviewer #2: Yes

3. Has the statistical analysis been performed appropriately and rigorously? 

Reviewer #1: Yes

Reviewer #2: Yes

4. Have the authors made all data underlying the findings in their manuscript fully available?

Reviewer #1: Yes

Reviewer #2: Yes

5. Is the manuscript presented in an intelligible fashion and written in standard English?

Reviewer #1: Yes

Reviewer #2: Yes

6. Review Comments to the Author

Reviewer #1: (No Response)

Reviewer #2: Manuscript could be accepted in its present form as the authors have adequately addressed my comments.

7. PLOS authors have the option to publish the peer review history of their article (what does this mean?). If published, this will include your full peer review and any attached files.

Reviewer #1: No

Reviewer #2: **Yes: **Dr. Afzal Hussain

---

## [Editor Report · Acceptance letter]

11 Jan 2022

PONE-D-21-13459R1 

Drivers of cadmium accumulation in *Theobroma cacao L.* beans: a quantitative synthesis of soil-plant relationships across the Cacao Belt 

Dear Dr. Wade:

I'm pleased to inform you that your manuscript has been deemed suitable for publication in PLOS ONE. Congratulations! Your manuscript is now with our production department. 

Kind regards, 

on behalf of

Dr. Muhammad Rizwan 

Academic Editor

PLOS ONE